# In situ coherent diffractive imaging

Yuan Hung Lo[1,2], Lingrong Zhao[1,3], Marcus Gallagher-Jones[1], Arjun Rana[1], Jared J. Lodico[1], Weikun Xiao[2], B.C. Regan[1] & Jianwei Miao[1]

Coherent diffractive imaging (CDI) has been widely applied in the physical and biological sciences using synchrotron radiation, X-ray free-electron laser, high harmonic generation, electrons, and optical lasers. One of CDI's important applications is to probe dynamic phenomena with high spatiotemporal resolution. Here, we report the development of a general in situ CDI method for real-time imaging of dynamic processes in solution. By introducing a time-invariant overlapping region as real-space constraint, we simultaneously reconstructed a time series of complex exit wave of dynamic processes with robust and fast convergence. We validated this method using optical laser experiments and numerical simulations with coherent X-rays. Our numerical simulations further indicated that in situ CDI can potentially reduce radiation dose by more than an order of magnitude relative to conventional CDI. With further development, we envision in situ CDI could be applied to probe a range of dynamic phenomena in the future.

[1] Department of Physics and Astronomy, and California NanoSystems Institute, University of California, Los Angeles, CA 90095, USA. [2] Department of Bioengineering, University of California, Los Angeles, CA 90095, USA. [3] Department of Physics and Astronomy, Shanghai Jiao Tong University, Shanghai 200240, China. Correspondence and requests for materials should be addressed to J.M. (email: miao@physics.ucla.edu)

The first experimental demonstration of coherent diffractive imaging (CDI) in 1999[1] has spawned a wealth of development in lensless imaging and computational microscopy methods with widespread scientific applications[2–32]. With continuous rapid development of coherent X-ray sources[33–36], high-speed detectors[37], and powerful algorithms[38,39], CDI methods are expected to have a larger impact across different disciplines in the future[36]. As many natural phenomena of interest evolve in response to external stimuli, CDI can make important contributions to the understanding of these dynamic phenomena[22,29,36,41,42].

Recently, in situ and operando X-ray microscopy have advanced to study dynamic processes with elemental and chemical specificity[43,44], but the spatial resolution is limited by the X-ray lens. While in situ electron microscopy can achieve much higher spatial resolution[45], the dynamic scattering effect limits the sample thickness and restricts the technique's applicability to a wider range of samples.

In this article, we demonstrate a general in situ CDI method to simultaneously reconstruct time-evolving complex exit waves of dynamic processes with spatial resolution only limited by diffraction signals. By introducing both static and dynamic regions in the experimental geometry, we apply the static region as a powerful time-invariant constraint to reconstruct dynamic processes with fast and robust convergence. Our numerical simulations indicate that with advanced synchrotron radiation, in situ CDI could potentially achieve 10 nm spatial resolution and 10 ms temporal resolution. Using an optical laser, we conduct proof-of-principle experiments of this method by capturing the growth of Pb dendrites on Pt electrodes immersed in an aqueous solution of $Pb(NO_3)_2$ and by reconstructing a time series of phase images of

live glioblastoma cells in culture medium. Furthermore, by varying the incident X-ray flux between the static and dynamic regions, we demonstrate through numerical simulations that in situ CDI can potentially reduce the radiation dose to radiation sensitive samples by more than an order of magnitude relative to conventional CDI.

## Results

**In situ CDI principle.** To achieve fast, reliable reconstruction of a time series of dynamic phenomena, in situ CDI takes advantage of two types of structures or regions. A dynamic region constantly changes over time or in response to external stimuli, while a static region remains stationary in time. A time series of far-field diffraction patterns are collected with interference between the static and dynamic regions. Since the static region remains unchanged during the data acquisition, this interference effectively creates a time-invariant overlapping region between the measured diffraction patterns, providing a powerful real-space constraint to simultaneously phase all diffraction patterns with fewer iterations and more robust convergence than conventional phase retrieval algorithms.

Figure 1a shows an experimental setup for in situ CDI. A dual-pinhole aperture is placed upstream of the sample to create two separate regions on the sample plane. The dynamic specimen of interest is localized to the area of one pinhole, while the other pinhole illuminates a region without the sample. Note that the second, static region can be completely empty or a substrate containing some stationary structure. Experimentally, the sample holder can be prepared by using microfluidics so that there are regions where one pinhole occupies the dynamic specimen while the other one covers a static area (Fig. 1a).

Furthermore, this technique can be used to extend scanning CDI techniques such as ptychography, where a region of interest can first be obtained by scanning, and then the dynamic specimen can be magnified and perturbed to probe dynamic information. As a general method, in situ CDI requires only a static region or structure between two consecutive time frames as the time-invariant constraint for phase retrieval, which can in principle be implemented with different experimental geometries.

**In situ CDI phase retrieval algorithm.** Figure 1b shows the schematic layout of the in situ CDI phase retrieval algorithm. Using random phases as an initial input, the algorithm iterates between real and reciprocal space with constraints incorporated in each space. The illumination function of the incident wave and a static function of the time-invariant overlapping region are enforced in real space, while the measured Fourier magnitudes are applied in reciprocal space. In each iteration, a weighted average static function is sequentially passed onto the reconstructions of the time series. Since the static function is shared and mutually reconstructed at different time frames, the solutions to the phase problem for the whole time series rapidly emerge without stagnation. The $j$th iteration of the algorithm consists of the following steps. Obtain a weighted average static function at time $t$,

$$S_{t,j'}(\mathbf{r}) = \gamma S_{t-1,j}(\mathbf{r}) + (1 - \gamma)S_{t,j}(\mathbf{r}) \qquad (1)$$

where $S_{t-1,j}(\mathbf{r})$ represents the static function at time $t-1$ and the weighting factor $\gamma$ is set to 0.8. Next, combine $S_{t,j'}(\mathbf{r})$ with a dynamic function, $D_{t,j}(\mathbf{r})$, to produce an object function,

$$O_{t,j}(\mathbf{r}) = S_{t,j}'(\mathbf{r}) + D_{t,j}(\mathbf{r}). \qquad (2)$$

Multiply the object function by the illumination function, $P(\mathbf{r})$, to

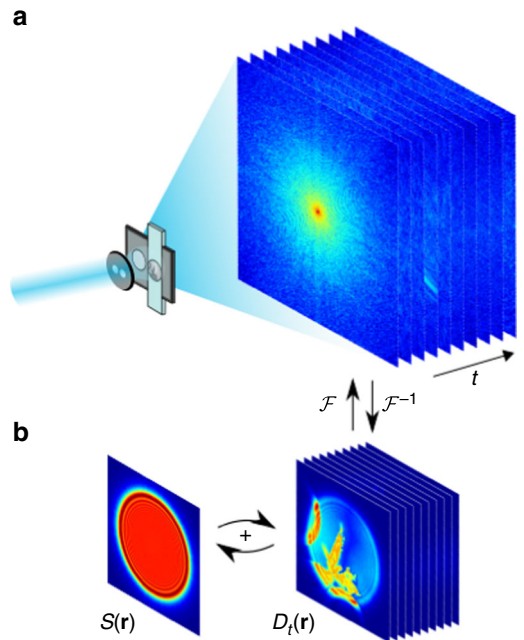

**Fig. 1** Schematic layout of the experimental geometry and the phase retrieval of in situ CDI. **a** A coherent wave illuminates a dual-pinhole aperture to create a static and a dynamic region, $S(\mathbf{r})$ and $D_t(\mathbf{r})$. A sample in the dynamic region changes its structure over time and a time series of diffraction patterns are collected by a detector. **b** By using the static region as a powerful time-invariant constraint in real space, the in situ CDI algorithm iterates between real and reciprocal space and simultaneously reconstructs a time series of complex exit waves of the dynamic processes in the sample with robust and fast convergence. $\mathscr{F}$ and $\mathscr{F}^{-1}$ represent the fast Fourier transform and its inverse, respectively

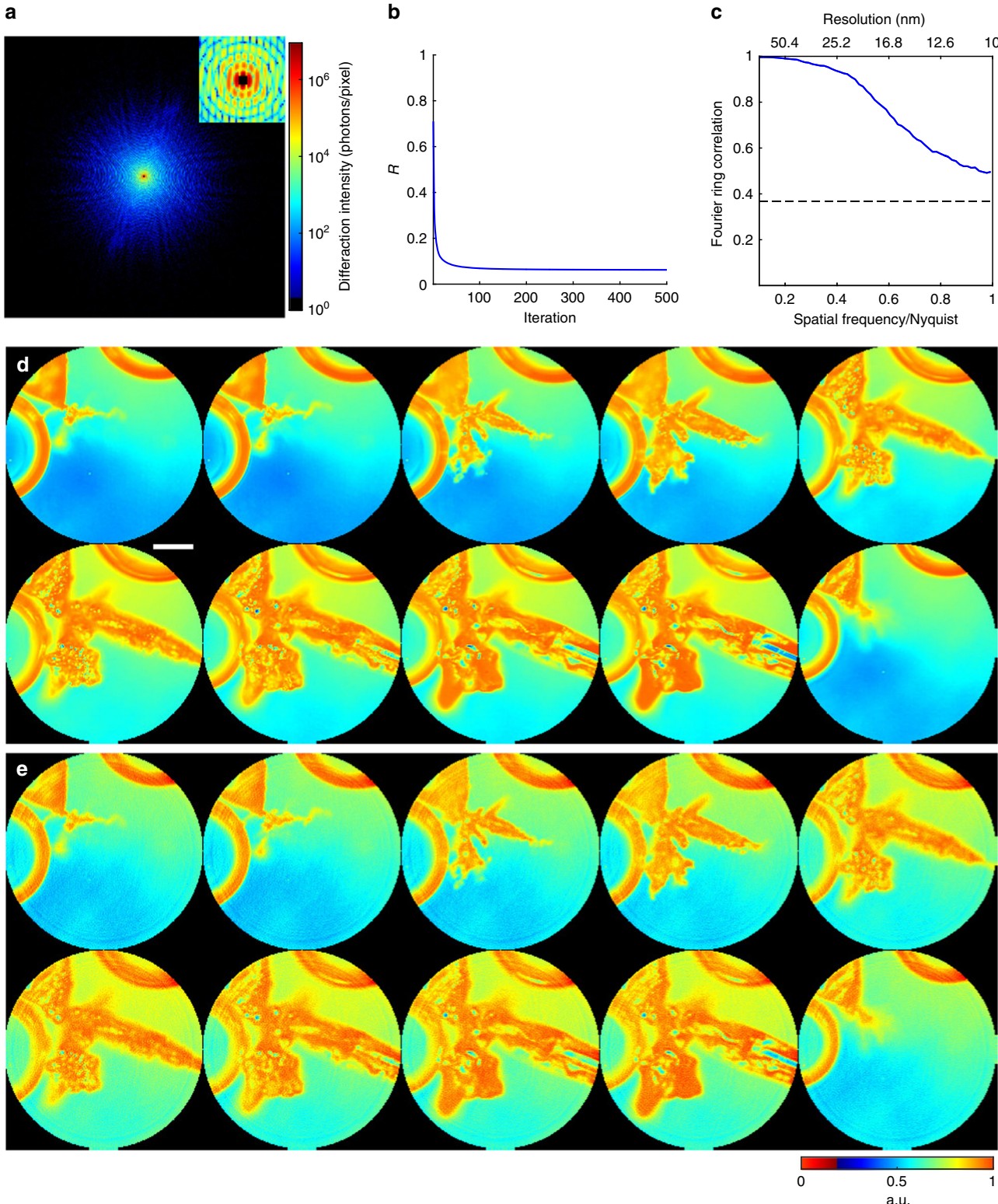

**Fig. 2** Numerical simulations of in situ CDI with coherent X-rays. **a** A representative diffraction pattern with Poisson noise and missing data, calculated from the Pb dendrite formation process in an electrochemical cell using 8 keV X-rays with a flux of $10^{11}$ photons $\mu m^{-2}$ s. The insert indicates a $5 \times 5$ pixels missing data at the center. **b** $R$-factor used to monitor the iterative algorithm, showing the rapid convergence of the algorithm. **c** Average Fourier ring correlation between a time-evolving structure model and its corresponding reconstructions indicates a spatial resolution of 10 nm can be achieved, with a temporal resolution of 10 ms. **d** The time-evolving structure model of the dendrite formation process immersed in a 1-μm-thick water layer. Scale bar, 200 nm. **e** The corresponding reconstructions of the time-evolving complex exit waves (showing only the magnitude), which are in good agreement with the structure model

generate a complex exit wave function,

$$\psi_{t,j}(\mathbf{r}) = O_{t,j}(\mathbf{r})P(\mathbf{r}). \qquad (3)$$

In the current version of the algorithm, an accurate knowledge of $P(\mathbf{r})$ is necessary, and can be experimentally measured[6,11,46]. Next, apply the fast Fourier transform (FFT) $\mathscr{F}$ to the exit wave function to obtain its Fourier transform,

$$\Psi_{t,j}(\mathbf{k}) = \mathscr{F}\left[\psi_{t,j}(\mathbf{r})\right]. \qquad (4)$$

and replace the calculated Fourier magnitude with the measured one,

$$\Psi'_{t,j}(\mathbf{k}) = \left|\Psi_t^m(\mathbf{k})\right| \frac{\Psi_{t,j}(\mathbf{k})}{\left|\Psi_{t,j}(\mathbf{k})\right|}. \qquad (5)$$

Apply the inverse FFT ($\mathscr{F}^{-1}$) to obtain an updated exit wave function,

$$\psi_{t,j'}(\mathbf{r}) = \mathscr{F}^{-1}[\Psi_{t,j}'(\mathbf{k})] \qquad (6)$$

then remove $P(\mathbf{r})$ to get an updated object function,

$$O_{t,j'}(\mathbf{r}) = O_{t,j}(\mathbf{r}) + \frac{|P(\mathbf{r})|P^*(\mathbf{r})}{\alpha(|P(\mathbf{r})|^2 + \varepsilon)}\left[\psi_{t,j}'(\mathbf{r}) - \psi_{t,j}(\mathbf{r})\right], \qquad (7)$$

where $\alpha = \max|P(\mathbf{r})|$ and $\varepsilon$ is a small value to prevent division by 0 [46]. Next, separate $O_{t,j}(\mathbf{r})$ into the updated static and dynamic functions, $S_{t+1,j}(\mathbf{r})$ and $D_{t,j+1}(\mathbf{r})$, respectively, and feed $S_{t+1,j}(\mathbf{r})$ back to beginning to reconstruct $D_{t+1,j}(\mathbf{r})$. After repeating above steps for the whole time series, $R$-factor is calculated for the $j$th iteration to monitor the convergence of the algorithm,

$$R_j = \frac{\sum_t \sum_{\mathbf{k}} \left| \left|\Psi_t^m(\mathbf{k})\right| - \left|\Psi_{t,j}(\mathbf{k})\right| \right|}{\sum_t \sum_{\mathbf{k}} \left|\Psi_t^m(\mathbf{k})\right|} \qquad (8)$$

After several hundred iterations, the algorithm quickly converges to the correct solution even in the presence of noise and missing data. Another unique feature of the algorithm is its ability to simultaneously reconstruct the complex exit waves of all frames without the necessity of averaging independent runs for individual frames.

**Numerical simulations of in situ CDI using coherent X-rays.** Batteries play an indispensable role in the development of modern technologies, but advances in high capacity batteries are hampered by dendritic growth, where microfibers of electrolyte materials sprout from the surface of electrodes during charge/discharge cycles and short the circuit. In some serious cases, dendrites can cause rapid heating and explosion of the battery[47]. While in situ transmission electron microscopy (TEM) can observe dendritic growth at high spatial resolution[48], the sample thickness is limited by the dynamical electron scattering effect and the temporal resolution is hampered by the electron flux[45]. Due to X-ray's larger penetration depth, in situ CDI is ideally suited to probe the dynamic phenomena of thick specimens with nanoscale spatial resolution and high temporal resolution.

To demonstrate in situ CDI's ability to reliably reconstruct dynamic structures, we performed numerical simulations on real time imaging of Pb dendrite growth in solution (Methods) (Fig. 2). Coherent X-rays with 8 keV energy and a flux of $10^{11}$ photons $\mu m^{-2} s^{-1}$ were incident on a dual-pinhole aperture. The illumination function was generated by propagating an exit wave from the dual-pinhole aperture to the sample plane. One of the pinholes illuminated the growth process of Pb dendrites immersed in a 1-$\mu$m-thick water layer (Methods), while the

other pinhole was focused on a static region. A time series of diffraction patterns were collected by a $1024 \times 1024$ pixel detector with a frame rate of 100 Hz and a linear oversampling ratio of ~2 [48]. Poisson noise was added to each diffraction pattern and the central $5 \times 5$ pixel data was removed to simulate the missing center problem (Fig. 2a).

By using random phase sets as an initial input, the in situ CDI algorithm quickly converged to the correct solution after several hundreds of iterations (Fig. 2b). Figure 2d shows a time series of the magnitudes of the reconstructed complex exit waves with a temporal resolution of 10 ms, which are in good agreement with the original structure model (Fig. 2c). Compared to conventional phase retrieval algorithms[38,39,50–52], the in situ CDI algorithm can produce very consistent final reconstructions with different random phase sets as the initial input. To quantify the reconstructions, we calculated the Fourier ring correlation (FRC, Methods) between the reconstructed images and the original structure models, indicating a spatial resolution of 10 nm was achieved in this case (Fig. 2b).

**Optical laser experiment with battery material.** As a proof-of-principle experiment, we demonstrated in situ CDI for materials science applications by capturing the growth of Pb dendrites on Pt electrodes immersed in an aqueous solution of Pb(NO$_3$)$_2$. A helium-neon (HeNe) laser was used as the coherent light source and illuminated a dual-pinhole aperture composed of two 100 $\mu$m holes spaced 100 $\mu$m apart edge-to-edge (Methods). An electrochemical cell was placed 400 $\mu$m downstream of the aperture. The cell was made from 50 $\mu$m diameter Pt wires immersed in 1.5 M Pb(NO$_3$)$_2$ solution and encased between two 100-$\mu$m-thick coverslips (Methods). The left pinhole was placed in front of the electrochemical cell, while the right pinhole was focused on the substrate devoid of any dendrite. Twelve DC voltages were applied to the electrochemical cell to generate Pb dendrite growth and dissolution. At each voltage, a diffraction pattern was measured by a liquid-nitrogen cooled CCD detector with $1340 \times 1300$ pixels and a pixel size of $20 \times 20$ $\mu$m. To validate the in situ CDI results, a $5 \times 5$ ptychographic scan was also collected at each voltage.>

Figure 3a and b and Supplementary Fig. 1 show the in situ CDI and ptychographic reconstructions of the same sample area at 12 different voltages. The overall structures are in good agreement between the two methods and the independent in situ CDI reconstructions are also very consistent (Supplementary Fig. 2). Figure 3c and d show some fine features are resolved in in situ CDI, but blurred in the ptychographic reconstruction. This blurring is due to continuous dendrite dissolution as the aperture scans over the field of view, resulting in an average reconstruction within the ptychographic scan.

Our results show that, as the voltage was ramped up to 1.8 V, Pb was rapidly deposited on the tip of the Pt wire to form short and wide dendrites (Supplementary Movie 1). Initially the growth continued as the voltage decreased to 1.5 V, but as the potential decreased further the dendrite began to dissolve from its tip down to the root. The dendrite did not fully dissolve from the tip during the measurement, even after the voltage was reversed. The presence of undissolved Pb dendrites increases the surface roughness of the electrode and can lead to enhanced dendrite growth in subsequent charge/discharge cycles. This highlights dendrite growth as a significant problem in rechargeable batteries, where many repeated charge/discharge cycles occur over the lifespan of the battery[47].

**Optical laser experiment with biological sample.** Tumor cell interaction offers insights into cancer progression, including recognition, communication, and assembly among cell groups[53]. Tumor cell fusion, or fusogenic events, has also been suggested as

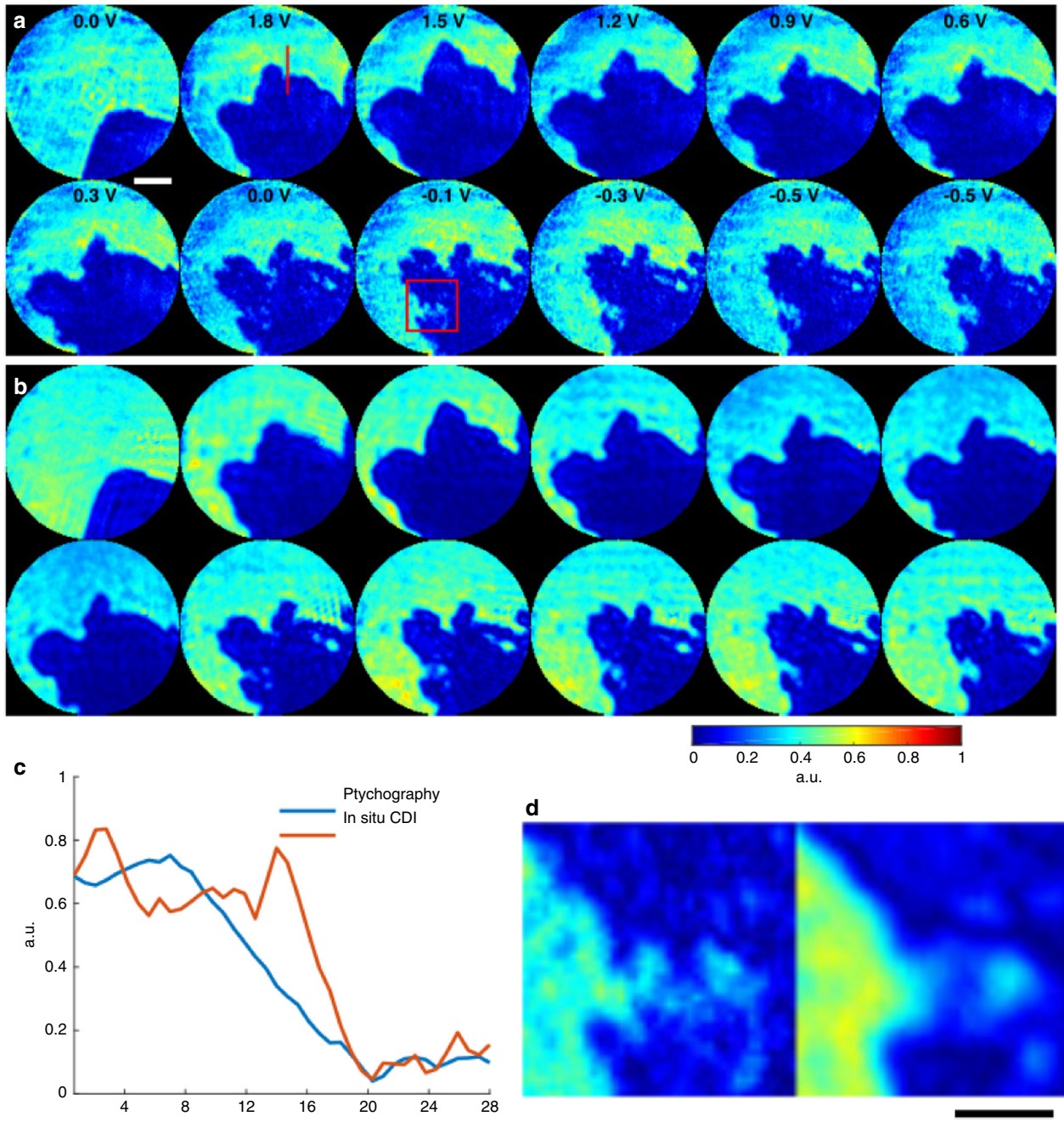

**Fig. 3** Proof-of-principle experiment on in situ CDI with a materials science sample. **a** The magnitude of the complex exit waves reconstructed by in situ CDI, capturing the growth of Pb dendrites on Pt electrodes immersed in an aqueous solution of Pb(NO₃)₂ as a function of the applied voltage. Scale bar, 20 μm. **b** Ptychographic reconstructions of the same dynamic sample area. The overall structures agree well between the two methods. However, some fine features are resolved in in situ CDI, but blurred in the ptychographic reconstruction as indicated by a line-out (**c**) and a magnified view (**d**) of two areas. The blurring in ptychography is due to the continuous dendrite dissolution as the aperture scans over the field of view. Scale bar, 10 μm

a source of genetic instability, as well as mechanisms for metastasis and drug resistance[54]. The fate of fused cells could be either reproduction or apoptosis, with still unclear implications. To demonstrate in situ CDI in a biological context, we used a HeNe laser and collected a time series of 48 diffraction patterns from live glioblastoma cells sealed between two cover slips (Methods). To validate our method for imaging the biological specimen, a 5 × 5 ptychographic scan was also collected at each time point.

Figure 4 and Supplementary Fig. 3 show a good agreement between the unwrapped phase images of in situ CDI and ptychographic reconstructions. The phase images show a small

cell, about 25 μm in length, slowly approaching and attaching to a larger cell, about 100 μm in length, over 2 h (Supplementary Movie 2). After 144 min, the large cell responded to the presence of the smaller cell and underwent a rapid morphological change. In the next three hours the large cell moved away from the small cell as the small cell's thin pseudopodium anchored and pulled on the large cell to keep it in place. In the subsequent 3 h, the two cells fused together and formed a dense circular shape. Another time series of the cells taken after fusion showed no noticeable morphological change or cell motility, suggesting that apoptosis occurred after the cells merged.

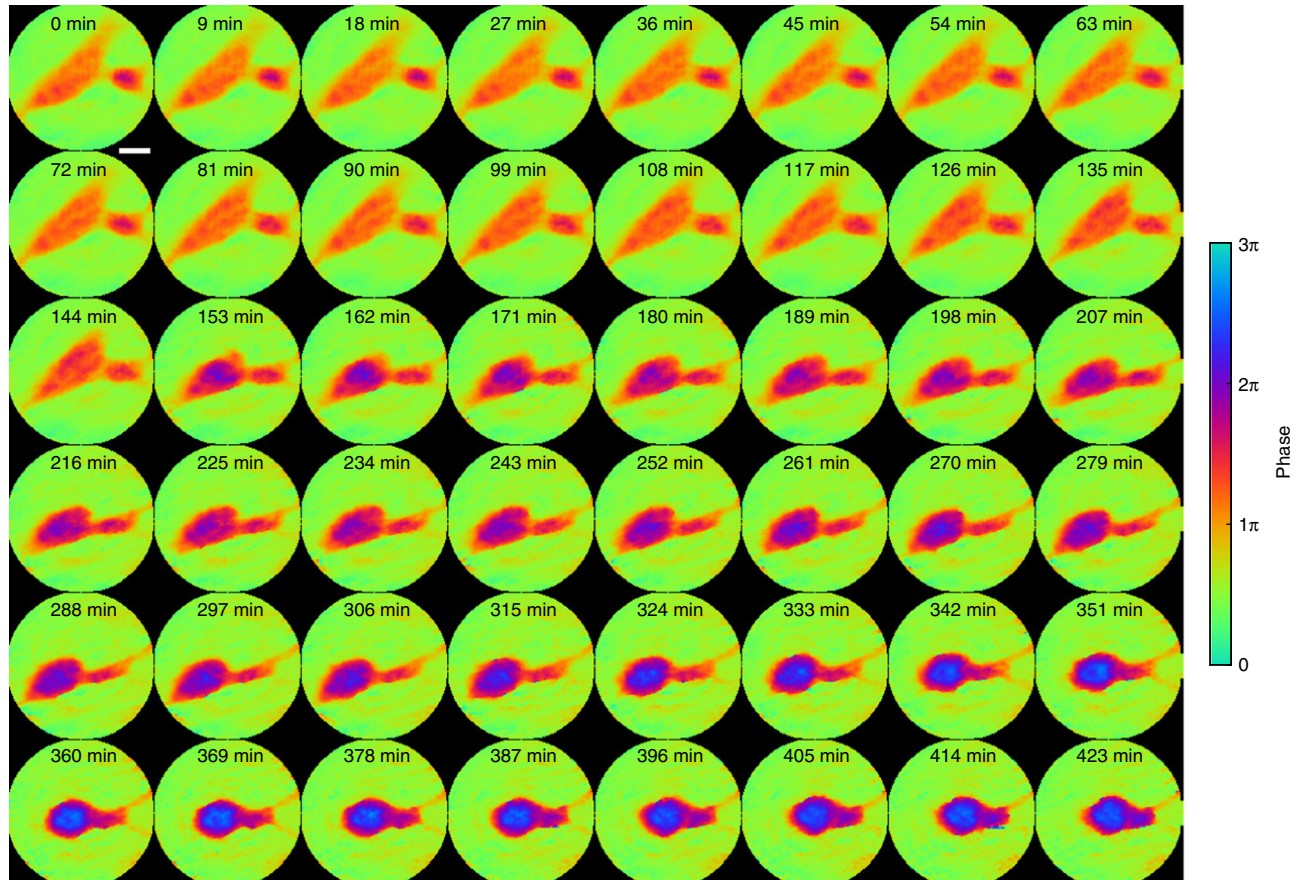

**Fig. 4** Proof-of-principle experiment on in situ CDI with a biological sample. Phase images of the fusion of glioblastoma cells reconstructed by in situ CDI. A smaller cell on the right approached a large cell and initiated cell attachment during the first 144 min. Upon attachment, the large cell underwent rapid morphology change and moved left, but the small cell anchored the large cell with thin pseudopodium on the right side of the field of view and began fusing until the 342nd min. The cells showed no motility post fusion, suggesting the occurrence of apoptosis following fusogenic event. Scale bar, 20 μm

**Numerical simulations of potential radiation dose reduction.** To image radiation-sensitive specimens with X-rays, the radiation damage process ultimately limits the achievable resolution[55,56]. One area currently being explored is the addition of a known diffusive structure to the sample to enhance the scattered signal. Placing a high atomic number element structure in the field of view has demonstrated the possibility of reducing the dose required for obtaining minimum reconstructable diffraction signal[57–62]. Since photons incident on the static region in in situ CDI do not hit the sample, those photons can enhance the measurable signal without inducing extra radiation damage to the sample. Furthermore, a carefully constructed static structure may also be used as additional a priori information to aid phase retrieval. Exploring a combination of these dose reduction strategies can help advance in situ CDI toward dynamic imaging of radiation-sensitive samples.

To examine the feasibility of dose reduction using an auxiliary scattering enhancing structure, we simulated a static structure of a 20-nm-thick Au pattern (Fig. 5a) and a biological sample consisting of a vesicle and protein aggregates (Fig. 5b, Table 1, Methods). Both the static structure and biological sample are submerged in 1-μm-thick $H_2O$ and masked by a 3 μm pinhole. Using coherent soft X-rays ($E = 530$ eV), we first calculated the diffraction patterns only from the biological sample with a total fluence varying from $3.5 \times 10^4$ to $3.5 \times 10^7$ photons μm$^{-2}$, corresponding to a radiation dose ranging from $2.75 \times 10^3$ to $2.75 \times 10^6$ Gy, respectively (Methods). The diffraction patterns were collected by a detector with quantum efficiency of 80% and Poisson noise was added to the diffraction intensity (Fig. 5c).

By using the oversampling smoothness (OSS) algorithm[52], we reconstructed the electron density of the biological sample from these noisy diffraction patterns (Fig. 5e–h). To quantify the spatial resolution, we calculated the Fourier ring correlation between the reconstructions and the model (Fig. 5m). Based on the $1/e$ criterion, we estimated the achieved resolution as a function of the total fluence.

Next, we calculated the diffraction patterns from a combination of the biological sample and the static structure. The total fluence on the biological sample varies from $3.5 \times 10^4$ to $3.5 \times 10^7$ photons μm$^{-2}$, while the total fluence on the static structure is fixed at $1.4 \times 10^{10}$ photons μm$^{-2}$. Experimentally, this can be implemented by introducing an absorber to the pinhole in front of the biological sample. The center-to-center distance between the biological sample and the static structure is 3.8 μm. Figure 5d shows the noisy diffraction pattern with a fluence of $3.5 \times 10^7$ photons μm$^{-2}$ on the sample and $1.4 \times 10^{10}$ photons μm$^{-2}$ on the static structure, which exhibits much higher spatial frequency diffraction signals than that calculated from only the biological sample with the same fluence (Fig. 5c).

By using the static structure as a constraint, we reconstructed the electron density of the biological sample from the noisy diffraction patterns, showing significant improvement in image quality and spatial resolution (Fig. 5i–l). According to Fourier ring correction (Fig. 5m, n), the in situ CDI method can reduce the radiation dose incident on the sample by more than an order of magnitude, while maintaining the same spatial resolution. In some cases (for example, $3.5 \times 10^4$ photons μm$^{-2}$ in Figs. 5m, n),

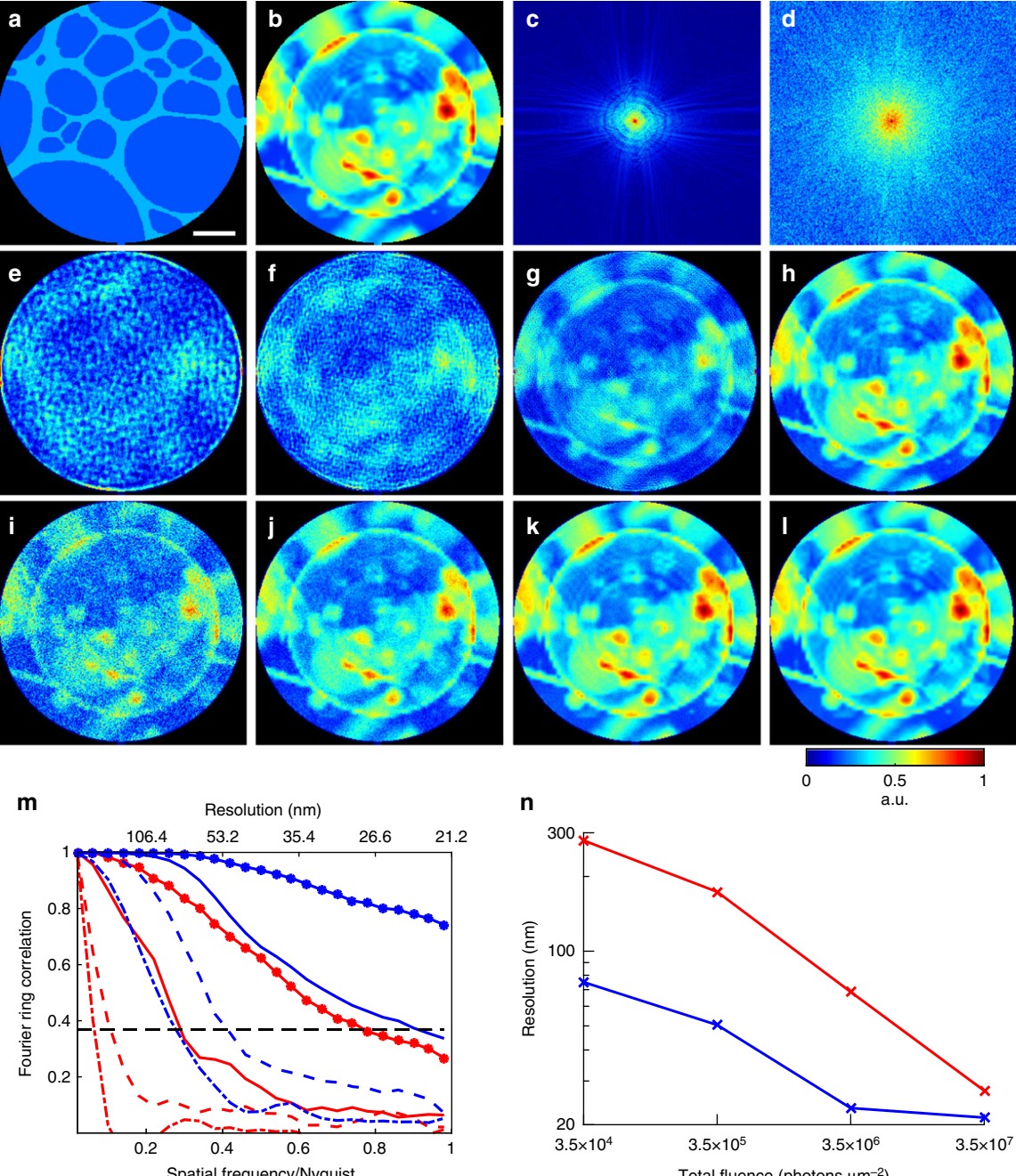

**Fig. 5** Numerical simulations of potential significant dose reduction using in situ CDI. **a** A simulated 20-nm-thick Au pattern in 1-µm-thick $H_2O$ static structure. The diameter of the pinhole is 3 µm. **b** A simulated biological sample of a vesicle and protein aggregates in 1-µm-thick $H_2O$. **c** Soft X-ray diffraction pattern with a 5 × 5 missing center, calculated from the biological sample with a photon energy of 530 eV and a fluence of $3.5 \times 10^7$ photons µm$^{-2}$. Poisson noise was added to the diffraction intensity. **d** Soft X-ray diffraction pattern calculated from the biological sample with a fluence of $3.5 \times 10^7$ photons µm$^{-2}$ and the static structure with a fluence of $1.4 \times 10^{10}$ photons µm$^{-2}$. Poisson noise was added to the diffraction intensity. The center-to-center distance between the biological sample and static structure is 3.8 µm. **e–h** Image reconstructions of the biological sample without the static structure, with fluences $3.5 \times 10^4$, $3.5 \times 10^5$, $3.5 \times 10^6$, and $3.5 \times 10^7$ photons µm$^{-2}$, respectively. **i–l** Image reconstructions with the same fluences on the biological sample as **e–h**, but with additional $1.4 \times 10^{10}$ photons µm$^{-2}$ on the static structure. **m** Fourier ring correlation of the reconstructions and the model. Red lines correspond to **e–h** (dash-dot, dashed, solid, solid-dotted lines, respectively), and blue lines to **i–l**. **n** Achieved spatial resolution of each reconstruction determined by the 1/e threshold in the Fourier ring correlation. Scale bar, 400 nm

the total dose can be reduced by two orders of magnitude with the same achievable resolution.

Our numerical simulations suggest that the level of radiation dose reduction is related to the structure of the static pattern and the ratio of the coherent fluxes between the static and dynamic structure. Although template-based and phase diverse approachs

have been introduced for radiation dose reduction in CDI[57,58], with in situ CDI we have demonstrated the possibility of reducing the radiation dose by more than an order of magnitude, which we attribute mainly to the non-linearity of the phase retrieval process. If experimentally validated, this could be the most dose efficient X-ray imaging method to probe radiation sensitive systems.

**Table 1 Dose reduction simulation parameters**

| | |
|---|---|
| *Simulation geometry* | |
| Detector size | 1100 × 1100 pixels |
| Detector quantum efficiency | 80% |
| Detector pixel size | 10 μm |
| X-ray energy | 530 eV |
| Sample-to-detector distance | 5 cm |
| Pinhole diameter | 3 μm |
| Fluence on dynamic structure | $3.5 \times 10^4$–$3.5 \times 10^7$ photons μm$^{-2}$ |
| Fluence on static structure | $1.4 \times 10^{10}$ photons μm$^{-2}$ |
| *Simulated sample parameters* | |
| Maximum dynamic structure thickness ($H_{50}C_{30}N_9O_{10}S_1$) | 1 μm |
| Static structure thickness (Au) | 20 nm |
| Protein density | 1.35 g cm$^{-3}$ |

## Discussion

In situ CDI overcomes a major challenge associated with traditional phase retrieval algorithms. In the presence of incomplete data and noise, conventional phase retrieval algorithms can be trapped in local minima and require averaging multiple independent runs to improve the final reconstruction[38,39,50–52]. By enforcing a time-invariant overlapping region as a powerful real-space constraint, in situ CDI is robust to incomplete data and noise, and it can simultaneously reconstruct a time series of complex exit waves to reveal fine structural changes between frames without being trapped in local minima.

Furthermore, the experimental configuration of in situ CDI can be improved by using a dedicated sample chamber, such as microfluidics (Fig. 1a), where the specimen of interest is physically separated from the static region by a barrier. Such customized experimental configuration could simplify the data collection and optimize the quality of reconstructions.

Compared to Fourier holography[63–65], in situ CDI has three unique distinctions. First, in Fourier holography, the spatial resolution is determined by the size of the reference source. In the X-ray regime, it is not only a challenge to fabricate very small reference sources, but also a small reference source would throw away a large fraction of coherent X-ray flux. On the other hand, in situ CDI does not have these limitations as its spatial resolution is only determined by the spatial frequency of the diffraction intensity.

Second, Fourier holography calculates the autocorrelation function from the hologram using the inverse Fourier transform[63–65]. To extract the image of a sample from its autocorrelation function, the sample and the reference source must satisfy a geometry requirement. But in situ CDI uses an iterative algorithm for phase retrieval and has no geometry requirement between the static and dynamic structure.

Third, in Fourier holography, the magnitude of the reference wave has to be comparable to that of the object wave for obtaining good quality autocorrelation functions. With in situ CDI, our numerical simulations indicate the coherent flux incident on the static and dynamic structure can vary by almost six orders of magnitude (Fig. 5i). Furthermore, by adjusting the coherent flux between the static and dynamic structure, one can potentially reduce the radiation dose to biological samples by more than an order of magnitude relative to conventional CDI (Fig. 5m).

For a proof-of-the-principle purpose, we assumed in the dose reduction simulations that the Au static structure was known a priori, which explains why single frame reconstructions can still be reliable achieved even without time series data. This is in contrast with the Pb dendrite simulation, where the static structure was not known a priori, and was instead reconstructed simultaneously with the dynamic structure, with the aid of a rough mask. The in situ CDI method is unique in that the redundant information does not need to be known explicitly, but can instead be obtained iteratively. To highlight the importance of time series data for the in situ CDI reconstruction, we ran the same 8 keV Pb simulation without knowing or enforcing the static structure. Supplementary Fig. 4 shows that the reconstructions without enforcing the static structure constraint have much lower quality than those with the static structure constraint enforced.

In essence, we have developed a general in situ CDI method for simultaneously reconstructing a time series of complex exit waves of dynamic processes. We validate this method using both numerical simulations and experiments on materials science and biological samples. Our numerical results indicate that the combination of in situ CDI and advanced synchrotron radiation can be used to image dynamic processes in solution with a spatial resolution of 10 nm and a temporal resolution of 10 ms. Using an optical laser, we have performed proof-of-principle experiments of in situ CDI by capturing the growth of Pb dendrites on Pt electrodes immersed in an aqueous solution of Pb(NO$_3$)$_2$ and reconstructing a time series of the phase images of live glioblastoma cells in culture medium.

There are four unique features associated with in situ CDI. First, it can simultaneously reconstruct a time series of complex exit waves with robust and fast convergence. Because no averaging is required in the reconstruction, fine structure variation at different time frames can be reliably reconstructed. Second, compared to liquid cell TEM[45], this method can be used to study the dynamics of a wider range of specimens (either thick or thin) in an ambient environment by optimizing X-ray energy based on the sample thickness and the chemical composition and reducing the multiple scattering effect.

Third, while ptychography uses partially overlapping structure in the space domain as a constraint, in situ CDI uses partially overlapping structure in the time domain as a constraint. Furthermore, by avoiding the requirement of sample scanning, in situ CDI can achieve higher temporal resolution than ptychography. Finally, this in situ approach can be applicable to any type of radiation with flexible experimental geometry as long as a static structure can be used as a time-invariant constraint. The spatial and temporal resolution of the method is ultimately limited by the coherent flux and the read-out time of the detector.

As coherent X-ray sources such as XFELs, advanced synchrotron radiation and high harmonic generation[33–36] as well as high-speed detectors[37] are under rapid development worldwide, we expect that this general in situ CDI method can potentially open the door to imaging a wide range of dynamical phenomena with high spatiotemporal resolution.

## Methods

**Numerical simulations of in situ CDI with coherent X-rays**. To generate a time-evolving structure model for the simulation, we scaled an optical microscopy video of Pb dendrites in an electrochemical cell (Supplementary Fig. 5). The thickest part of the Pb dendrites is 500 nm and the thickness of the water layer is 1 μm. Using the complex atomic scattering factor of Pb and H$_2$O at 8 keV[66], we calculated the projected complex electron density of the structure model as a function of time, $O_t(\mathbf{r})$. Next, we created a dual-pinhole aperture consisting of two 1-μm-diameter holes spaced 1.25 μm apart center-to-center. The dual-pinhole illumination function $P(\mathbf{r})$ was calculated by propagating the aperture function to the sample plane with a distance of 10 μm. Small random fluctuation is added to $P(\mathbf{r})$ to introduce the effect of imperfect illumination function estimate. The diffraction pattern at frame $t$, $I_t(\mathbf{k})$, was collected by a 1024 × 1024 pixel detector,

$$I_t(\mathbf{k}) = I_0 \eta \Delta t \left( \frac{r_e \lambda}{a \sigma_1} \right)^2 \left| \Psi_{D,t}(\mathbf{k}) + \Psi_S(\mathbf{k}) \right|^2 \tag{9}$$

where $\Psi_{D,t}(\mathbf{k})$ and $\Psi_S(\mathbf{k})$ are the structure factors of the dynamic and static functions at frame $t$, respectively. $\Psi_{D,t}(\mathbf{k}) + \Psi_S(\mathbf{k})$ was calculated by using the FFT as $\mathcal{F}[P(\mathbf{r}) \cdot O_t(\mathbf{r})]$, $I_0$ is the incident photon flux ($10^{11}$ ph $\mu m^{-2} s^{-1}$), $\eta$ is the detector efficiency (0.8), $\Delta t$ is the acquisition time (10 ms), $r_e$ is the classical electron radius, $\lambda$ is the wavelength, $\alpha$ is the size of illuminated area (3 μm), and $\sigma_1$ is the linear oversampling ratio[52] (2). To simulate the dynamic process, we calculated each diffraction pattern by integrating 10 individual patterns, each generated from an image in Supplementary Fig. 5 with 1 ms exposure. Poisson noise was added to the integrated diffraction patterns with a temporal resolution of 10 ms. This process introduces motion blurring that is more representative of experimental measurements (Supplementary Fig. 5).

**Experiment setup with a HeNe laser.** Our proof-of-principle experiments used a 543 nm HeNe laser (REO) with a power of 5 mW. A collimated beam with a diameter of 800 μm was directed onto a dual pinhole aperture, which consists of two 100 μm pinholes spaced 100 μm apart from edge to edge. The illumination was incident onto the sample 400 μm downstream of the aperture. A 35 mm objective lens was placed immediately downstream of the sample, and far-field diffraction patterns were measured by a $1340 \times 1300$ pixel CCD detector (16 bits, Princeton Instruments) at the lens' back focal plane. In order to increase the dynamic range of the diffraction intensity, three separate exposure times, 100, 1000, and 10,000 ms, were taken and the diffraction patterns were computationally stitched together without missing centers.

**Electrochemical cell preparation.** A sealed fluid cell was assembled to observe the dynamics of Pb dendrites. With the aid of an optical microscope, two platinum wires (diameter = 50.8 μm, 99.95% Alfa Aesar) were immersed in a thin layer of a saturated solution of $Pb(NO_3)_2$ (99.5%, SPI-Chem) in deionized water and were encapsulated between two glass microscope coverslips ($22 \times 22 \times 0.13$ mm$^3$). The two glass slides were epoxied together with the platinum wires exposed for making electrical contact.

**Glioblastoma cell preparation.** The glioblastoma cell line U-87 MG was purchased from ATCC (Manassas, Virginia). Cells were cultured in T75 cell culture flask (Thermo Fisher) in Dulbecco's Modified Eagle Medium (DMEM) (Thermo Fisher) with 10% fetal bovine serum (Corning Inc.) and 100 U/ml penicillin–streptomycin (Thermo Fisher) in 37 °C and 5% $CO_2$ incubator. To seed the cells onto coverslips, the cells were treated with TrypLE (Thermo Fisher) for 5 min in a 37 °C incubator. The reaction was stopped by adding an equal volume of the complete culture medium. Around 1 million cells were seeded in a 100 mm glass plate with four pieces of coverslips inside the plate to allow attachment.

**Potential significant radiation dose reduction using in situ CDI.** The dose reduction simulation used 530 eV X-rays to minimize water background and to ensure good cellular contrast. A dual-pinhole aperture with 3 μm diameter pinholes spaced 4 μm apart was used to illuminate the static structure and a biological sample covering $7 \times 7$ μm field of view. A $1100 \times 1100$ pixel detector with 10 μm detector pixel size and a $5 \times 5$ pixel missing center was placed 5 cm downstream of sample, with maximum resolution at detector edge of 21.2 nm and an oversampling ratio ($\sigma_1$) of ~2 ($\sigma_1$ of about 4 for single pinhole case). The simulated biological specimen consists of a 2 μm long organelle and various cytosolic components in a $3 \times 3$ μm$^2$ region of a 1 μm thick cell. The complex electron density of the biological specimen is calculated using the average composition of protein ($H_{50}C_{30}N_9O_{10}S_1$). Adjacent to the specimen is the static structure, composed of 20 nm thick Au pattern resembling a lacey carbon morphology. The recorded diffraction intensity $I(\mathbf{k})$ with 1 s exposure ($\Delta t$) was calculated as

$$I(\mathbf{k}) = \eta\Delta t\left(\frac{r_e\lambda}{a\sigma_1}\right)^2\{I_D|\Psi_D(\mathbf{k})|^2 + I_S|\Psi_S(\mathbf{k})|^2 + \sqrt{I_D I_s}\left(\Psi_D(\mathbf{k})\Psi_S^*(\mathbf{k}) + \Psi_D^*(\mathbf{k})\Psi_S(\mathbf{k})\right)\} \quad (10)$$

Where $\Psi_D(\mathbf{k})$ and $\Psi_S(\mathbf{k})$ are the complex waves of the biological specimen and static structure, respectively, calculated using tabulated atomic scattering factors of their respective materials. $I_S$ and $I_D$ are photon fluxes on the static structure and biological specimen, respectively. Eq. (10) is an expansion of Eq. (9) to allow for differential flux through each structure. In the case that $I_D$ equals $I_S$, Eq. (10) is reduced to Eq. (9).

Phase retrieval on the simulated noisy diffraction intensity was performed using OSS[46] with 500 iterations. The reconstruction with the lowest Fourier R-factor in 10 independent runs was used as the final result. Resolution was quantified by the Fourier ring correlation (FRC),

$$FRC(\mathbf{k}) = \frac{\sum \Psi_m(\mathbf{k}) \cdot \Psi_g(\mathbf{k})}{\sqrt{\sum |\Psi_m(\mathbf{k})|^2 \cdot \sum |\Psi_g(\mathbf{k})|^2}} \quad (11)$$

where $\Psi_m(\mathbf{k})$ and $\Psi_g(\mathbf{k})$ are the complex structure factors of the model and reconstruction, respectively.

**Quantification of radiation dose in simulation.** In the simulation, we estimated the radiation doses (D) imparted on the biological specimen as[55,56],

$$D = \left(\frac{P_t}{A}\right)\left(\frac{\mu E}{\rho}\right) \quad (12)$$

where total incident X-ray photons ($P_t$) per unit area ($A$) through a 3 μm pinhole ($P_t/A$) varies from $3.5 \times 10^4$ to $3.5 \times 10^7$ photons $\mu m^{-2}$. The cell density ($\rho$) is 1.35 g cm$^{-3}$, and the linear absorption coefficient ($\mu$) of average protein at 530 eV photon energy (E) is $1.25 \times 10^4$ cm$^{-1}$, which gives a mass absorption coefficient ($\mu/\rho$) of $9.26 \times 10^3$ cm$^2$ g$^{-1}$. Thus, the total dose delivered to the biological specimen ranges from $2.75 \times 10^3$ to $2.75 \times 10^6$ Gy.

**Data availability.** The MATLAB source code of the in situ CDI algorithm and data that support the findings in this study are freely available from http://www.physics.ucla.edu/research/imaging/dataSoftware.html.

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

## Acknowledgements

We thank D.K. Toomre for a stimulating discussion about the in situ CDI idea. This work was supported by STROBE: A National Science Foundation Science & Technology Center under Grant No. DMR 1548924 and the DARPA PULSE program through a grant from AMRDEC. J.M. acknowledges the partial support by the Office of Basic Energy Sciences of the US DOE (DE-SC0010378).

## Author contributions

J.M. conceived the idea and directed the research; Y.H.L., L.Z., M.G.-J., A.R., J.J.L., W.X., B.C.R., and J.M. designed and/or conducted the experiments. Y.H.L., L.Z., M.G.-J., A.R., and J.M. analyzed the data, performed reconstructions, and interpreted the results. Y.H.L. and J.M. wrote the manuscript with contributions from all authors.

## Additional information

**Competing interests:** The authors declare no competing interests.

