## [Peer Review File · Nature Communications]

Reviewers' comments:

Reviewer #1 (Remarks to the Author):

The paper by Lo et al presents an algorithm for performing simultaneous phase retrieval on a time series of diffraction patterns from a dynamic structure. Two demonstration experiments are performed with visible light illumination and simulations are provided to characterize expected performance with coherent x-rays. The reconstruction algorithm is novel and does appear to work in practice but several of the claims in the paper are not substantiated by the provided evidence and I do not see sufficient novelty to warrant publication in *Nature Communications*. I recommend submission in a more specialized journal.

A few technical comments:

1. At a few places throughout the manuscript, the authors state their method enjoys "robust and fast" convergence. There is no data provided to satisfy this claim. It is not clear how algorithm robustness would be quantified but speed certainly can. There is no data presented about the algorithm speed. There is only one analysis (Fig 2B) of the fidelity of a single reconstruction. Statements about algorithm speed and robustness should be removed if further analysis is not added.
2. The authors go to great lengths to argue that this method is unique from Fourier Holography. Indeed, the reconstruction method is different but the experimental geometry is not, except for a restriction on the separation required for holographic reconstruction. This method is more general in terms of the allowable geometry but the required illumination scheme and the data generation are not novel.
3. I do not believe the x-ray simulations add anything meaningful to this paper. As far as I can tell, the data are generated from static structures and presented as reconstructions of a dynamic system. For instance, a single diffraction pattern in the time series is $|\text{FFT}\{P(x)*(S(X) + D(X,T))\}|^2$ where S is the static portion of the sample and D is the dynamic portion of the sample. However, how are the dynamics of D included in the simulation? In the equation above, $D(x,T)$ is completely independent of the dwell time dT . How would the diffraction pattern change in this scenario for longer dwell times, except by improving statistics? If the system were really dynamic, longer dwell times would blur out the diffraction pattern similar to the effects of partial coherence. Are these considerations taken into account? If not, all claims about achievable spatial and time resolution are not accurate.

Reviewer #2 (Remarks to the Author):

The authors have responded to comments but have not addressed fundamental problems with the manuscript as previously raised.

The question that was previously asked and remains unanswered is exactly how can the results of the experiment and the simulation therefore be complementary if they are performed with entirely different optical conditions that require an entirely different optical setup? For this work to warrant publication in any reputable journal, it will need to reconcile the simulations that were supposedly performed at 8 keV with real experiments performed under comparable conditions. Such an approach is quite normal evidenced based scientific practice.

The authors claim that visible light experiments "demonstrate that this methodology can be used to image the dynamic process of both materials science and biological (weak phase) samples in solution" and that "resolution is not the focus of this demonstration". In a contradictory manner, the authors then go on to state a method for achieving high spatial resolution. This is rather concerning.

Without including results from experiments performed under comparable conditions, the authors have not demonstrated broad applicability of the method in the very fields of study where impact is claimed. Achieving sufficient resolution under experimental conditions is without doubt crucial in demonstrating that the method can yield meaningful results at 8 keV and that there are no practical inhibitions along the way.

The work in its current state is an incomplete study and therefore does not warrant publication.

Reviewer #3 (Remarks to the Author):

As a reviewer to a previous draft of this article, many of my concerns seem to have been addressed in this updated draft. Some specific comments follow:

- Line 70: The authors say that this can be used for extended samples, but they need a pinhole with no sample behind it for this technique. I don't see how this method could be used to image anything but the edges of a time-varying extended sample without destroying it.
- Minor point, but Fig. 2b should be split into two plots. This much optimization of space usage is unnecessary and reduces clarity.

The next few points are regarding the dose tolerance simulations, which although quite interesting, do raise some questions:

- The simulation is just static, which is similar to the cited references, especially ref. 57, except for the spatial separation of the reference. This link should be more strongly acknowledged.
- The static simulation also raises the question of why one needs the whole time-series of data in the first part of the paper if one had a known reference and knew the beam profile a priori. The authors should show a comparison of reconstruction quality of a single frame independently vs with the whole data set.

- Is there a missing beamstop region in these simulations? If not, why?

- Most of the resolution enhancement between conventional and in situ CDI is due to the more reliable phasing, rather than due to dose efficiency, which is at most a factor of 2 due to the holographic reference. If the strong reference produces ψ_{ref} and the sample is ψ_{sam} ,

CDI: $|\psi_{\text{sam}}|^2$

in situ CDI: $|\psi_{\text{ref}}|^2 + 2 \psi_{\text{ref}} \psi_{\text{sam}}$

This point should be emphasized.

In conclusion, the updated manuscript is interesting and mostly well supported by the data shown (except for the minor points raised above). With minor changes, I recommend that this paper be published in Nature Communications.

Reviewer #4 (Remarks to the Author):

I think that the authors have successfully addressed all the comments in my previous report. In the revised manuscript, radiation dose reduction using in situ CDI is supported by their newly-performed numerical simulation. Also, fundamental differences between in situ CDI and Fourier holography is

now clarified in response to other referees' comment.

Related to comment 6 from referee 1, I have questions on the separation of O into S and D. Does the circle in Supplementary Fig. 1d represent the mask or the illumination? Please add the circles indicating the illumination and the mask, and the scale bar to the figure. Please also describe the applied voltage for the data shown in Supplementary Fig. 1d and 1e.

Response to the referees:

We thank four referees for reviewing our revised manuscript and making good suggestions. We have made a second revision to our manuscript and fully addressed all the points raised by the referees. Below is a point-by-point response to the referees' comments and suggestions.

Response to referee 1:

Comment 1: The reconstruction algorithm is novel and does appear to work in practice but several of the claims in the paper are not substantiated by the provided evidence and I do not see sufficient novelty to warrant publication in Nature Communications. I recommend submission in a more specialized journal.

We respectfully disagree with this point. Please see our reply to Comment 3 as well as our reply to referee 2 in detail.

Comment 2. At a few places throughout the manuscript, the authors state their method enjoys "robust and fast" convergence. There is no data provided to satisfy this claim. It is not clear how algorithm robustness would be quantified but speed certainly can. There is no data presented about the algorithm speed. There is only one analysis (Fig 2B) of the fidelity of a single reconstruction. Statements about algorithm speed and robustness should be removed if further analysis is not added.

This is a good point. To address the claim of robust convergence, we have included further analysis in Supplementary Fig. 2. We performed 10 independent *in situ* CDI reconstructions using the Pb dendrite experimental dataset, each starting with random initial guesses. We plotted two line scans through the edge and interior of the dendrite and included the standard deviations to show small variations among the reconstructions (Supplementary Fig. 2b). We also included the variance map of all 10 reconstructions to demonstrate the result's consistency and the robustness of the algorithm (Supplementary Fig. 2c). As for the claim of fast convergence, we meant that the algorithm reaches a minimum with a smaller number of iterations (less than 100) than traditional phase retrieval algorithms (Fig. 2b), which can be trapped in local minima and can take a few thousand iterations to reach the convergence. We have clarified this point in the revision (line 61). Finally, to facilitate those who may be interested in our work, we have committed to post all the source codes of *in situ* CDI and all the data on a public website (www.physics.ucla.edu/research/imaging/dataSoftware.html) immediately after this manuscript is published online.

Comment 3. The authors go to great lengths to argue that this method is unique from Fourier Holography. Indeed, the reconstruction method is different but the experimental geometry is not, except for a restriction on the separation required for holographic reconstruction. This method is more general in terms of the allowable geometry but the required illumination scheme and the data generation are not novel.

We respectfully disagree with this point. There are four differences between *in situ* CDI and Fourier holography in terms of the experimental geometry, required illumination and data generation. First, as referee 1 has correctly pointed out, Fourier holography has a restriction on

the separation between the sample and the reference source, but *in situ* CDI does not have such a restriction. Second, in Fourier holography, the spatial resolution is determined by the size of the reference source. Thus the experimental geometry used for *in situ* CDI will not work for Fourier holography. Third, with the *in situ* CDI geometry, our numerical simulations indicate the coherent flux incident on the static and dynamic structure can vary by almost six orders of magnitude (Fig. 5), allowing radiation dose reduction by more than an order of magnitude relative to conventional CDI. In contrast, in Fourier holography the magnitude of the reference wave has to be comparable to that of the object wave for obtaining good quality autocorrelation functions. Finally, while we used a dual-pinhole geometry in this work, *in situ* CDI can in principle be implemented to any type of radiation with flexible experimental geometry as long as a static structure can be used as a time-invariant constraint. This is why we believe that the experimental geometry, required illumination and data generation of *in situ* CDI are novel enough to warrant its publication in Nature Communications.

Comment 4. I do not believe the x-ray simulations add anything meaningful to this paper. As far as I can tell, the data are generated from static structures and presented as reconstructions of a dynamic system. For instance, a single diffraction pattern in the time series is $|\text{FFT}\{P(x)(S(X) + D(X,T))\}|^2$ where S is the static portion of the sample and D is the dynamic portion of the sample. However, how are the dynamics of D included in the simulation? In the equation above, $D(x,T)$ is completely independent of the dwell time dT . How would the diffraction pattern change in this scenario for longer dwell times, except by improving statistics? If the system were really dynamic, longer dwell times would blur out the diffraction pattern similar to the effects of partial coherence. Are these considerations taken into account? If not, all claims about achievable spatial and time resolution are not accurate.*

Following referee 1's suggestion, we have incorporated more realistic dynamic simulations in the revision. We first generated a dynamic structure model, consisting of 100 frames with 1 ms per image (Supplementary Fig. 5). Using a flux of 10^{11} photons/ $\mu\text{m}^2/\text{s}$, we calculated each diffraction pattern by integrating 10 individual patterns, each generated from a 1 ms image. This process introduces motion blurring that is more representative of experimental measurements (Supplementary Fig. 5). After including missing data and adding Poisson noise to each diffraction pattern with a temporal resolution of 10 ms, we applied *in situ* CDI to reconstruct the dynamic structures. Figure 2d shows the time-evolving structure model with a temporal resolution of 10 ms, each of which was averaged from ten 1-ms images (Supplementary Fig. 5). Figure 2e shows the corresponding reconstructed images, which are in good agreement with the model. We have clarified these points in our revision (see Fig. 2, Supplementary Fig. 5 and Methods in detail).

Response to referee 2:

Comment 1. The question that was previously asked and remains unanswered is exactly how can the results of the experiment and the simulation therefore be complementary if they are performed with entirely different optical conditions that require an entirely different optical setup? For this work to warrant publication in any reputable journal, it will need to reconcile the simulations that were supposedly performed at 8 keV with real experiments performed under comparable conditions. Such an approach is quite normal evidenced based scientific practice.

The authors claim that visible light experiments "demonstrate that this methodology can be used to image the dynamic process of both materials science and biological (weak phase) samples in solution" and that "resolution is not the focus of this demonstration". In a contradictory manner, the authors then go on to state a method for achieving high spatial resolution. This is rather concerning.

Without including results from experiments performed under comparable conditions, the authors have not demonstrated broad applicability of the method in the very fields of study where impact is claimed. Achieving sufficient resolution under experimental conditions is without doubt crucial in demonstrating that the method can yield meaningful results at 8 keV and that there are no practical inhibitions along the way.

We strongly, but respectfully, disagree with these points. As we described in our previous response, conceptually novel papers are usually more important than those with sophisticated experiments. For example, the first holography paper [D. Gabor, Nature 161, 777-778 (1948)] used an optical beam to demonstrate the recording of amplitudes and phases in one diagram, aiming to correct the spherical aberration of electron lenses. Holography was later demonstrated using electron microscopy by others. But the 1971 Nobel Prize was awarded to Gabor. Also, the first CDI paper used soft X-rays to demonstrate the measurement and reconstruction of the diffraction pattern from a non-crystalline test pattern with 75 nm resolution [J. Miao et al. Nature 400, 342-344 (1999)]. This resolution was lower than that achievable with conventional X-ray microscopy at that time. According to referee 2's opinions, these classic papers would have been rejected because the Gabor paper did not demonstrate holography with electrons and the first CDI paper did not achieve better resolution than conventional X-ray microscopy.

In this work, we used an optical laser to validate the principle of *in situ* CDI by capturing the growth of Pb dendrites on Pt electrodes in an aqueous solution and by reconstructing a time series of the phase images of live glioblastoma cells in culture medium. We further confirmed the method by performing numerical simulations with coherent X-rays. Moreover, by adjusting the coherent flux between the static and dynamic structure, we demonstrated through numerical simulations the possibility of reducing the total dose to radiation sensitive samples by more than an order of magnitude relative to conventional CDI. To our knowledge, this could be the most dose efficient X-ray imaging method to probe radiation sensitive systems. This is why we believe this work is conceptually novel enough to warrant publication in Nature Communications.

Response to referee 3:

Comment 1: Line 70: The authors say that this can be used for extended samples, but they need a pinhole with no sample behind it for this technique. I don't see how this method could be used to image anything but the edges of a time-varying extended sample without destroying it.

We want to clarify this point. In our *in situ* CDI experiments of imaging the Pb dendrite growth and live glioblastoma cells, we first performed a ptychographic scan to locate a dynamic and a less-dynamic regions and then placed one pinhole on the dynamic region and the other on the near static region. In another words, our experimental geometry does not require complete empty behind the pinholes. Furthermore, while we adopted a dual-pinhole geometry in this work, *in situ* CDI can in principle be implemented to image extended objects using flexible experimental geometry as long as a static structure can be used as a time-invariant constraint. For example, we are designing a microfluidic sample chamber for *in situ* CDI, where the specimen of interest is

physically separated from the static region by a barrier. Such customized experimental configuration could simplify the data collection and optimize the quality of reconstructions. Also, we are developing another sample chamber using pre-designed, high-Z structure as the static region. Our numerical simulations have indicated that such a design works well for *in situ* CDI. We have clarified this point in the revision.

Comment 2: Minor point, but Fig. 2b should be split into two plots. This much optimization of space usage is unnecessary and reduces clarity.

Following referee 3's suggestion, we have split Fig. 2b into Fig. 2b and 2c in the revision.

Comment 3: The next few points are regarding the dose tolerance simulations, which although quite interesting, do raise some questions: The simulation is just static, which is similar to the cited references, especially ref. 57, except for the spatial separation of the reference. This link should be more strongly acknowledged.

Following referee 3's suggestion, we have clarified this point and acknowledged the connection with ref. 57 on lines 249-254.

Comment 4: The static simulation also raises the question of why one needs the whole time-series of data in the first part of the paper if one had a known reference and knew the beam profile a priori. The authors should show a comparison of reconstruction quality of a single frame independently vs with the whole data set.

This is a very good point. If the static structure is known *a priori*, as is the case in the dose reduction simulation, then single frame reconstructions indeed works well in reconstructions. However, in both the experimental reconstructions and the 8 keV numerical simulation we assumed we do not know the static structure *a priori*, and only supplied a rough mask that is slightly bigger than the area of the static structure to crop and feed through the reconstructed static structure (in this case, the second pinhole). So in effect we reconstructed both dynamic and static structures simultaneously. To emphasize this point, we included an additional result of the 8 keV numerical simulations, where we did not enforce the static structure constraint between frames and let all time series of data to reconstruct in parallel. Supplementary Fig. 4 shows that the reconstructions without enforcing the static structure constraint have lower quality than those using the static structure constraint. We have clarified this important point in the revision (lines 288-299).

Comment 5: Is there a missing beamstop region in these simulations? If not, why?

In our previous simulations, we assumed a 10- μm beamstop (1 pixel) to block the direct beam. To make the simulation more realistic, we re-ran the simulations with a 5x5 pixel (50- μm) beamstop, and similar results were obtained (see updated Fig. 5 in detail).

Comment 5: Most of the resolution enhancement between conventional and in situ CDI is due to the more reliable phasing, rather than due to dose efficiency, which is at most a factor of 2 due

*to the holographic reference. If the strong reference produces ψ_{ref} and the sample is ψ_{sam} ,
CDI: $|\psi_{sam}|^2$
in situ CDI: $|\psi_{ref}|^2 + 2\psi_{ref}\psi_{sam}$
This point should be emphasized.*

This is an important point. Our numerical simulations have indicated that, by adjusting the coherent flux between the static and dynamic structure, one can potentially reduce the dose to radiation sensitive samples by more than an order of magnitude relative to conventional CDI (Fig. 5). This is more than ‘a factor of 2 due to the holographic reference’ suggested by referee 3. We have performed more numerical simulations using different structures and obtained consistent results. We attribute this potential significant dose reduction to the non-linearity of the phase retrieval process. To our knowledge, this could be the most dose efficient X-ray imaging method to probe radiation sensitive systems. We have clarified this point in the revision (lines 249-254).

Response to referee 4:

Comment 1: Related to comment 6 from referee 1, I have questions on the separation of O into S and D. Does the circle in Supplementary Fig. 1d represent the mask or the illumination? Please add the circles indicating the illumination and the mask, and the scale bar to the figure. Please also describe the applied voltage for the data shown in Supplementary Fig. 1d and 1e.

Following referee 4’s suggestions, we have added solid and dashed circles to indicate the illumination and the mask, and also added a scale bar and the applied voltage in the revised Supplementary Fig. 1.

REVIEWERS' COMMENTS:

Reviewer #3 (Remarks to the Author):

The referee response and updated manuscript addresses my comments. I now recommend this version for publication.

Reviewer #4 (Remarks to the Author):

I think the authors addressed all the comments raised by me and other referees. I have no further comments on the revised manuscript. In conclusion, I think the paper is now suitable for publication in Nature Communications.